# Ophthalmological Manifestations in People with HIV from Northeastern Romania

**DOI:** 10.3390/medicina59091605

**Published:** 2023-09-06

**Authors:** Mihaela Cobaschi, Isabela Ioana Loghin, Victor Daniel Dorobăț, George Silvaș, Șerban Alin Rusu, Vlad Hârtie, Victoria Aramă

**Affiliations:** 1Faculty of Medicine/Clinical II Department, “Carol Davila” University of Medicine and Pharmacy, 050474 Bucharest, Romania; cobaschimihaela@gmail.com (M.C.); victordorobat@yahoo.com (V.D.D.); dr.arama@mateibals.ro (V.A.); 2National Institute for Infectious Diseases “Prof. Dr. Matei Balș”, 021105 Bucharest, Romania; 3Department of Infectious Diseases, “Grigore T. Popa” University of Medicine and Pharmacy, 700115 Iasi, Romania; vladhartie@yahoo.com; 4Department of Infectious Diseases, “St. Parascheva” Clinical Hospital of Infectious Diseases, 700116 Iasi, Romania; silvas.george@gmail.com (G.S.); rususerbanalin@yahoo.com (Ș.A.R.); 5Department of Intensive Care, University Hospital of Emergency, 050098 Bucharest, Romania; 6Department of Intensive Care, Clinical Hospital of Emergency “Prof. Dr. Nicolae Oblu”, 700309 Iasi, Romania

**Keywords:** ophthalmological diseases, HIV infection, CMV retinitis

## Abstract

*Background and Objectives*: Although ocular disorders can occasionally impact people with HIV over the course of their illness, HIV/AIDS is unmistakably a multisystem disorder. A physician can rule out a wide range of ophthalmic problems with the assistance of an ophthalmologist, from adnexal disorders to posterior segment diseases, including those affecting the optic tract and optic nerve. *Materials and Methods*: Based on patient medical data from the “St. Parascheva” Clinical Hospital of Infectious Diseases in Iasi, we carried out a retrospective clinical investigation on patients with HIV/AIDS and ophthalmological conditions who were hospitalized in northeastern Romania. We seek to draw attention to the characteristics and ophthalmological comorbidities of HIV/AIDS patients. The studied period was between 1 January 1991 and 31 December 2022. *Results*: There were a total of 38 recorded cases of ophthalmological manifestations in the HIV-infected patients. The research group’s average age was 37.31 years old (standard deviation 9.5693917). Males were primarily impacted, having lower total CD4+ T-lymphocyte levels based on sex and CD4+ T-lymphocyte levels overall. The HIV viral load was 999 268.13 copies/mL on average (standard deviation 1,653,722.9). Of all the patients, we found out that 17 had congenital eye diseases (44.73%) and the others (21, 55.26%) developed ophthalmological diseases. CMV Retinitis was found most frequently, in eight patients (21.05%), followed by Myopia in seven patients (18.42%). *Conclusions*: The key to the management of HIV-positive patients is a multidisciplinary approach and access to antiretroviral therapy. Anyone who is HIV-positive and experiences ocular symptoms at any time should be directed to seek professional ophthalmologic treatment as soon as feasible. A therapeutic holdup could result in irreversible vision loss. Long-term coordination is required to combat this disease, improving communication between the ophthalmology and infectious disease fields.

## 1. Introduction

The immune system is the target of the human immunodeficiency virus (HIV) infection. Acquired immunodeficiency syndrome (AIDS) is one of the illness’s most severe manifestations. HIV weakens the immune system by targeting the body’s CD4 T lymphocytes. This makes contracting illnesses such as tuberculosis, infections, and some malignancies easier [1,2].

HIV/AIDS is clearly a multisystem disorder, but ophthalmic diseases do sometimes affect HIV-positive individuals throughout the natural course of their disease. HIV may damage the eye either directly or indirectly through a number of opportunistic illnesses. Herpes zoster ophthalmicus, cotton wool patches in the retina (the most frequent ophthalmoscopic finding in HIV-infected people), and molluscum contagiosum affecting the eyelids are ocular symptoms indicating towards an HIV infection in an undiagnosed and otherwise asymptomatic patient [3,4]. With the help of an ophthalmologist, a doctor can rule out a variety of ocular issues before starting antiretroviral treatment (ART), from adnexal illnesses to posterior segment diseases, including those affecting the optic tract and optic nerve [5,6].

Adnexa diseases include exfoliative dermatitis, such as Steven–Johnson syndrome, conjunctival molluscum contagiosum, and Herpes Zoster ophthalmicus. Syphilis, uveitis, Herpes keratitis, and candidal keratitis are among the anterior segment illnesses. The non-infectious posterior segment symptoms of HIV include cotton wool patches, hemorrhages, telangiectasias, and optic disc atrophy [7,8].

Acute retinal necrosis, Candida endophthalmitis, tubercular choroiditis, Cryptococcal or Pneumocystis choroiditis, and Cytomegalovirus ocular disease are some of the infectious posterior eye segment diseases [9,10].

In contrast to adults with a lower incidence of CMV retinitis, children are more prone to experiencing neurodevelopmental delays, keratitis sicca, and ocular toxoplasmosis [11]. The presence of blue sclera, hypertelorism, many palpebral fissures, and downward obliquity of the eyes are additional indicators of fetal AIDS-related embryopathy [12]

Ophthalmologists can help with the diagnosis and treatment of other eye diseases associated with HIV infection, such as the implantation of an intraocular ganciclovir implant, which can deliver higher intraocular ganciclovir levels than systemic therapy alone and lowers the risk of a CMV ocular disease relapse. Furthermore, surgically reattaching a retinal detachment caused by CMV can partially restore the eyesight in HIV-positive patients. Today, this condition’s prevalence has decreased due to the introduction of antiretroviral medication [13,14].

Adverse effects from drugs can also affect the eyes. Rifabutin is one such medication that might cause uveitis, especially when coupled with azole antifungal medications. The antiviral cidofovir, used in some cases of CMV retinitis, is also linked to uveitis and a drop in intraocular pressure [15].

Immune reconstitution inflammatory syndrome must be understood by ophthalmologists today, since it might result in uveitis after the start of ART. Fundoscopy is essential for the follow-up evaluation of patients using ART, since this condition affects the vitreous, macula, and optic nerve and sporadically causes cataracts and epiretinal membranes [16].

When people with HIV complain of any visual issue, it is strongly advised that they be referred to an ophthalmologist. The eyes of patients with HIV should undergo a thorough examination. Health education about the problems and visual symptoms associated with HIV will raise awareness and lower morbidity. The early detection and rapid treatment of these ocular symptoms will help to avoid or lessen the effects of subsequent vision deterioration [17,18].

Ocular symptoms should be ruled out for people with HIV in a discussion between an infectious disease physician and an ophthalmologist. Untreated ophthalmic issues spread quickly and have poor prognoses. Because of vision loss and opportunistic infections of the retina, all HIV-positive patients should be advised to undergo a baseline ophthalmologic examination. To avoid permanent vision loss, all HIV-infected individuals who have ocular symptoms must receive ophthalmologic care as soon as possible [19,20].

## 2. Materials and Methods

### 2.1. Study Design and Database Information

On the foundation of hospital medical data, we carried out a retrospective clinical investigation of patients diagnosed with HIV/AIDS in the northeastern region of Romania, hospitalized in “Sf. Parascheva” Clinical Hospital of Infectious Diseases in Iasi, aiming to emphasize the profile and ophthalmological-associated comorbidities of HIV/AIDS cases. The studied period was between 1 January 1991 and 31 December 2022.

Patients over the age of 18 who were hospitalized at our Regional HIV/AIDS Center in northeastern Romania and tested as HIV-positive via an enzyme-linked immunosorbent assay (ELISA) test, confirmed to have the disease by Western blot (WB), were chosen for inclusion. The HIV plasma viral load and CD4+ T cell counts were also assessed in the patients who tested positive for HIV/AIDS. In our study group, 38 patients were included.

### 2.2. Ethical Approval

The “St. Parascheva” Clinical Hospital of Infectious Diseases in Iasi, Romania, gave the study its clearance. (May 2023; Approval No. 4/17). At the time of admission, every individual signed a waiver of informed consent.

### 2.3. Study Variables

Age- and gender-specific demographic information, personal pathological histories, clinical traits, blood tests (viro-immunological testing), assessments of potential opportunistic infections, patient staging, the start of antiretroviral treatment, and the course and outlook of the patients with HIV/AIDS infection were all included in the data collection.

According to the Center for Disease Control and Prevention (CDC), Atlanta, the HIV infection stage was determined using an age-specific CD4+ T-lymphocyte count or the CD4+ T-lymphocyte percentage of the total CD4 T-lymphocyte cells level. HIV infection and AIDS are classified into three stages: stage 1, when CD4+ T-lymphocyte levels are above 500 cells/μL; stage 2, when they are between 200 and 499 cells/μL; and stage 3, when they are below 200 cells/μL. HIV infection is represented by stages 1 and 2, while AIDS is represented by stage 3 [4,6].

Two ELISA tests were used to serologically assess those suspected of having HIV, and a Western blot test was used to confirm the diagnosis. The regional public health management network’s epidemiologists carried out all of this work, after which, the patients were sent to the local HIV/AIDS facility.

### 2.4. Study Setting

The “St. Parascheva” Clinical Hospital of Infectious Disease, Iasi, is a primary referral medical facility for the Moldova region of Romania, with a capacity of 300 beds. It is divided into six pavilions. Pavilion V includes a compartment of Infectious Disease and the HIV/AIDS Regional Center. The Regional Center has a capacity of 12 beds, where patients are periodically evaluated based on the CDC and EACS recommendations.

The hospital’s central laboratory completed all the blood tests, and the molecular biology lab measured the patients’ CD4+ T cell counts and HIV plasmatic viral loads. RT-PCR HIV 1 was utilized with Cepheid’s GeneXpert^®^ as a tool for measuring the viral load levels and determining the HIV viremia. If the viral load was less than 40 copies/mL, it was deemed to be undetectable, and when it was greater than 40 copies/mL, it was deemed to be detectable.

Periodically, the clinical and biological status of every newly diagnosed PWH (person living with HIV) was assessed for metabolic syndrome and liver enzymes. The laboratory reference values were within 5–31 UI/L for AST (aspartate trans-aminase) and ALT (alanine transaminase), within 7–32 UI/L for GGT (gamma-glutamyl transferase), within 122–200 mg/dL for COL (cholesterol), within 30–159 mg/dL for LDL-COL (low-density lipoprotein cholesterol), 40–66 mg/dL for HDL-COL (high-density lipoprotein cholesterol), and 30–150 mg/dL for TG (triglycerides), with no differences between sexes.

The Regional HIV/AIDS Center, Iasi, has a total of 1692 patients in its active records. They are periodically evaluated every six months to ensure their adherence and compliance with the antiretroviral treatment. Each patient has a medical file that mentions their associated diseases, blood test values, level of CD4 T lymphocytes, and HIV viral load, as well as the ART schemes they followed. The data in this study were collected from the paper charts of the patients of our center.

### 2.5. Statistical Analysis

The Pearson test in the XLSTAT version 2019 program was used to determine the correlation between the demographic characteristics, clinical data, and results. Kendall’s Tau correlation coefficients were established [11]. The Statistical Software for Excel (XLSTAT) version 2019 was used to conduct the statistical analysis.

## 3. Results

In the northeastern part of Romania, from a total of 1692 patients in the active records, there were a total of 38 recorded cases of ophthalmological manifestations in HIV-infected patients. Men were the ones who experienced these most frequently (21 cases, 55.26%) compared to women (17 cases, 44.74%). In total, 17 HIV-infected patients had congenital eye diseases and the other 21 developed ophthalmological diseases (Figure 1).

Most of the instances involved young adults, aged between 30 and 39 years old—28 patients (73.68%), next, the age group of 40–49 had 4 patients (10.53%), 50–59 years—2 patients (5.26%), over 60 years old—2 patients (5.26%), 0–19 years—1 patient (2.63%), and 20–29 years—1 patient (2.63%) (Table 1). The study group’s median age was 37.31 years old.

Nearly half of the patients in our research group were from Iasi, according to the distribution of the group by county (17 cases, 44.74%), next, Suceava (7 cases, 18.42%), Botosani (5 cases, 13.16%), Neamt (3 cases, 7.89%), Bacau (3 cases, 7.89%), and Vaslui (3 cases, 7.89%), (Figure 2). There were 26 patients who came from the urban region of northeast Romania (68.42%), and the other 66 cases (31.58%) were rural residents.

Given the route of transmission, each case indicated a potential cause. The most common was the perinatal route, with a total of 24 cases (63.16%).

The most afflicted group in terms of the sexual method of transmission (heterosexual and MSM/men having sex with men—34.21% instances) was young adult males (aged 21–40) with a medium level of education (graduated high school). Drug usage administered intravenously was noted in 2.63% of patients.

The virological and immunological status of the study group of patients from the Iasi HIV/AIDS Regional Center was assessed. With an average CD4+ T-lymphocyte level of 372.34 cells/μL (standard deviation 292.60658), it was seen that 36.84% of cases had a CD4+ T-lymphocyte level of 1–199 cells/μL, 31.58% of cases had a CD4+ T-lymphocyte value of 200–499 cells/μL, and 31.58% of cases had a CD4+ T-lymphocyte value over 500 cells/μL (Table 2, Figure 3). Males were primarily impacted, having lower total CD4+ T-lymphocyte levels based on sex and CD4+ T-lymphocyte levels. The HIV viral load was assessed to be 999,268.13 copies/mL on average (standard deviation 1653722.9).

The following findings were obtained using the CDC (Center for Disease Control and Prevention) phases of HIV/AIDS: according to our data, 12 patients (31.58%) had a stage 1 HIV infection, 12 patients (31.58%) had a stage 2 HIV infection, and 14 patients (36.84%) had a stage 3 HIV infection.

In the research group, it was found that nearly a third of the males (18.42%) and less than a third (13.15%) of the females (13.15%) had abnormal ALT and AST values, correspondingly. In terms of metabolic profile, the triglyceride levels were abnormal in a third of the participants in the study group, impacting nearly equally each sex (21.05% men and 18.42% females), and the blood cholesterol levels were elevated in a little over half of the sample, independently of sex (26.31% males and 13.15% females). (Table 3). Regarding the metabolic and liver panels, further studies need to focus more on the impact of enzymes on eye manifestations in people with HIV or AIDS.

The research group underwent testing for the most prevalent co-infections linked to HIV/AIDS. The findings revealed that various opportunistic infections affected more than half (55.26%) of the patients admitted to our clinic throughout the study period. When the CD4 T-lymphocyte count fell below 500 cells/μL in stages 2 and 3 of infection (15.79% and 31.58%, respectively), several opportunistic infections were discovered. (Table 4).

The findings revealed that hepatitis B co-infections were the most common (21.05% of patients), *Cytomegalic virus* (CMV) (18.42% of patients), followed by hepatitis C (5.26% of patients) and toxoplasmosis (5.26% of patients). Males had a greater incidence of co-infections than females, making them more afflicted. The Pearson test in the XLSTAT version 2019 program was used to determine a *p*-value of 0.048 (Table 5).

In all the patients, we found out that 17 had congenital eye diseases (44.73%) and the others (21, 55.26%) developed ophthalmological diseases (Table 6). CMV Retinitis was found in eight patients (21.05%), Myopia in seven patients (18.42%), Convergent strabismus in three patients (7.89%), Astigmatism in four patients (10.52%), and cataracts in five patients (13.15%) (Table 7). Other diseases such as retinal detachment (3 patients, 7.89%), Molluscum contagiosum infection of the eyelids (2 patients, 5.26%), periorbital abscesses (2 patients, 5.26%), and acute conjunctivitis (4, 10.52%) were found in 11 patients (28.94%).

The specific treatment of the ophthalmological conditions identified in the study patients was instituted in collaboration with an ophthalmologist. It should be mentioned that an antiviral treatment with Gancyclovir 5 gm/kgc bid IV for 3 weeks was used in cases of CMV retinitis, in accordance with the specialist guidelines. In total, 28.94% benefited from optical correction and for those with cataracts, surgical treatment was used.

Antiretroviral therapy (ART) was prescribed to every patient at the Iasi HIV/AIDS Regional Center who were confirmed. Therefore, in 18.42% of cases, a single drug was administered, and in the remaining 81.57% of cases, a medication regimen that addressed the patient’s comorbidities was prescribed. The period of time between diagnosis and ART initiation ranged between 72 h and 14 days, according to the severity of the cases, in order to avoid IRIS (immune reconstruction inflammatory syndrome) also following www.hiv-druginteractions.org (accessed on 1 January 2022).

The most used antiretroviral regimen was protease inhibitors (17 patients, 44.73%), then a regimen based on nucleoside reverse transcriptase inhibitors (NRTIs) with non-nucleoside reverse transcriptase inhibitors (NNRTIs) being used for 10 patients (26.31%), and lastly, a regimen based on integrase inhibitors (9 patients, 23.68%). Other antiretroviral regimens included CCR5 inhibitors used for two patients (5.26%) (Table 8).

Following a one-month evaluation of the patients, the viro-immunological state revealed an elevated level of CD4+ T cells and marked decline in HIV viremia. Due to this, 15 patients (39.47%) had a CD4 value between 200 and 499 cells/μL, 8 patients (21.05%) had a value under 200 cells/μL, and 15 patients (39.47%) had a value over 500 cells/L. Most patients of both sexes had a CD4+ T-lymphocyte count of 200–499 or more cells/μL (Table 9).

The patients were assessed at the time of diagnosis and again a month after beginning ART. After beginning antiretroviral medication, the HIV viral load significantly decreased, with viral suppression occurring in 55.26% of cases (21 instances). The patients who were undetectable at the initial assessment were either transferred from a different regional HIV/AIDS center or diagnosed abroad and had already started ART when first evaluated at our Regional HIV/AIDS Center (Table 10).

## 4. Discussion

According to the Infectious Disease Society of America, people with HIV who have a CD4 level under 50 cells are recommended to receive ophthalmological care.

This study evaluated the frequency of eye conditions in HAART-treated people with HIV. The participants’ ages ranged from 18 to 70 years old, with a mean age of 38.64 years and an SD of 12.84 years. In total, 71% of the participants were between the ages of 26 and 55. In total, 40% of the patients were female and 60% were male. The ratio of men to women was 1.5.

Bekele S. et al. found that the prevalence of ocular symptoms was 25.3% overall within 348 patients (175 were on antiretroviral therapy and 173 were not on therapy). Keratoconjunctivitis sicca (11.3%) was the most prevalent ocular manifestation, followed by blepharitis (3.2%), molluscum contagiosum (2.6%), conjunctival squamous cell carcinoma (2.3%), conjunctival micro vasculopathy (2.3%), cranial nerve palsies (2%), herpes zoster ophthalmicus (HZO) (1.2%), and HIV retinopathy (0.6%). Patients with a CD4+ level of less than 200 cells/L frequently developed HIV retinopathy and conjunctival micro vasculopathy, whereas patients with a CD4+ count of 200–499 cells/L frequently developed HZO and molluscum contagiosum. Patients on HAART had a higher prevalence of ocular manifestations (32.6%) than non-HAART patients (17.9%) [21]. Our study found more cases with a CD4 level of 200–499 cells/mL who already had ophthalmic congenital diseases or developed other infectious eye diseases. CMV retinitis was found in cases with a CD4 level of <200/mL.

Li W. et al. found 667 (8%) individuals with ocular disorders among the 8743 hospitalized HIV/AIDS patients (15 116 cases) that were enrolled in their study. Ocular injuries were found in 65 (2%) of the 2902 patients who underwent a non-professional inspection, 46 (2%) of the 1621 patients who underwent an on-demand inspection, and 556 (13%) of the 3553 patients who underwent a routine check. The majority of HIV/AIDS ocular manifestations (354 [53%] of 667 patients) were caused by infectious diseases. Most infections affected the cornea, conjunctiva (152 [43%] of 354 patients), and retina (145 [41%] of 354 patients), respectively. Retinopathy and retinitis were strongly linked with CD4-positive counts of fewer than 200 cells/mL [22]. Our study found that, besides infectious eye diseases, many patients had chronic congenital ophthalmic diseases such as astigmatism, myopia, and convergent strabismus.

Gurung S. et al. investigated 54 children and 60 adults, which made up 114 cases, and 24.9% of the children and 61.9% of the adults showed ocular signs. Disorders of the anterior segment and external ocular system made up 21% of the cases in this research. Herpes simplex blepharoconjunctivitis (11.1%) was found in seven children, and dry eye (8.3%) and herpes zoster ophthalmicus (2.6%) were observed in adults, and were the most prevalent findings. Adult instances of posterior segment symptoms (HIV retinopathy 13.5%, CMV retinitis 10.8%, retinal detachment 8.1%, multifocal choroiditis 2.7%, and ocular toxoplasmosis 2.7%) made up 34% of the patients in this study [23]. In our study, we did not include children with infectious eye diseases, but we had similar results in terms of adult cases.

Hothi H. et al. observed that, in patients with HIV who were taking HAART, the prevalence of ocular manifestations was 39%. Adnexal involvement, anterior segment involvement, posterior segment involvement, neuro-ophthalmic abnormalities, and orbital involvement all occurred in 20%, 28%, 33%, and 4% of these cases, respectively. In total, 51% of patients had CD4+ T cell counts under 200 cells/L, and 76% were in the WHO clinical stages 2 and 3 [24]. We observed that infectious eye diseases were more frequent in cases with CD4 levels of <200/mL and 201–499/mL.

Rekha K.R. et al. observed that the overall prevalence of ocular manifestations was 23%, 9.3% of which involved the anterior segment and 26.6% involved the posterior region [25]. We found that the prevalence of anterior segment diseases was more frequent (42.10%), followed by posterior segment manifestations (28.94%) and adnexa diseases (28.98%).

In HIV/AIDS patients, Di Y. et al. observed that CMVR was the most prevalent ocular complication, followed by uveitis and HIV retinopathy. The percentage of patients who had CMVR and HIV retinopathy remained constant over time, however, the percentage of patients who had uveitis showed a clear upward trend (from 1.14% to 19.32%) [26]. The most frequent disease in our study was CMV retinitis (8, 21.05%), followed by myopia (7, 18.42%).

Mustapha J et al. studied a total of 103 patients with HIV. A total of 51.5% of the research participants reported impaired visual acuity in at least one eye, and 44.7% had at least one ocular problem. The most prevalent disorders were toxoplasmic retinochoroiditis (3.9%), posterior vitreous detachment (2.9%), blepharitis (10.7%), nucleosclerosis (6.8%), conjunctivitis (5.8%), pinguecula (5.8%), and dry eye (21.4%) [27].

According to Becker et al., postmortem investigations indicate that the rates of ocular findings are closer to 90%, and that between 50 and 70 percent of people with HIV eventually acquire ocular manifestations. CMV retinitis and retinal microvasculopathy are the two visual signs of AIDS that occur most often, although there are numerous additional problems that might result in vision loss. In order to minimize HIV replication, HAART employs a mix of antiretroviral medications; as a result, the CD4+ helper T-cell population typically recovers, leading to a decrease in opportunistic infections, an improvement in quality of life, and decreases in morbidity and death. This has been demonstrated by a 50% or greater decline in the incidence rates of CMV retinitis and other ocular diseases linked to HIV/AIDS [28]. Besides HAART therapy, patients need frequent ophthalmological evaluations to ensure an increased quality of life, furthermore if they have associated eye diseases.

Yuan TH et al. stated that metabolism, oxidative stress, and inflammation are only a few of the pathways that illustrate the intimate relationship between the liver and the eye. Future research on the connection between the liver and the eye will help us to better understand the communication mechanism between the two organs, which will improve our understanding of the pathogenesis and progression of these liver or eye diseases and help us to develop new therapeutic targets and more effective clinical treatments [29].

This study has potential limitations. The model is based on the retrospective observational nature of the study. The limitations of this paper include its small sample size and retrospective nature. In addition, it is possible that other patients were admitted for territorial ophthalmological care.

Nevertheless, HIV is still a serious public health concern. Infected people can experience the virus’s rapid mutation. Due to this ability, HIV has been able to evolve a variety of defenses against the body’s immunological responses and antiviral treatment. However, it is hoped that ongoing research may result in newer, low-toxicity anti-HIV drugs that more efficiently lower the viral loads of infected patients and stop them from spreading the illness to other people. By collaborating with other physicians in primary care and lending support to international organizations that promote HIV testing for patients at risk, offer pre-test counseling, and enhance access to low-cost antiretroviral treatment, ophthalmologists can help to prolong the lives of infected patients and reduce the risk of virus transmission.

It is important to ensure the long-term follow-up of HIV-infected patients who present ophthalmic manifestations at the national level, so that we can diagnose and ensure rapid treatment in order to increase quality of life.

## 5. Conclusions

Despite adequate antiretroviral coverage, HIV infection is still identified in our country after a considerable amount of time. Priority must be given to encouraging voluntary testing, especially among those most at risk for infection (men who have sex with other men, people who use drugs, young adults under 24 years old, sex workers, and those from impoverished backgrounds) [30].

Additionally, anyone with HIV who experiences ocular symptoms should seek specialized ophthalmologic treatment as soon as possible. Any therapeutic holdup could result in irreversible vision loss. Long-term coordination is required to combat this terrible disease, improving communication between the ophthalmology and infectious disease fields [31].

The longevity of patients with HIV is expanding due to the widespread use of HAART and development of better drugs. Additionally, they are more likely to experience sight loss and more likely to acquire ocular symptoms. A multidisciplinary team is best suited to addressing HIV, because it affects numerous organs. When individuals with HIV complain of any visual issue, it is strongly advised that they be referred to an ophthalmologist. The eyes of those with HIV should undergo a thorough examination [32].

## Figures and Tables

**Figure 1 medicina-59-01605-f001:**
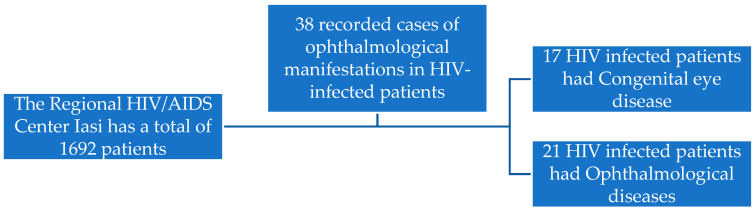
Study case findings.

**Figure 2 medicina-59-01605-f002:**
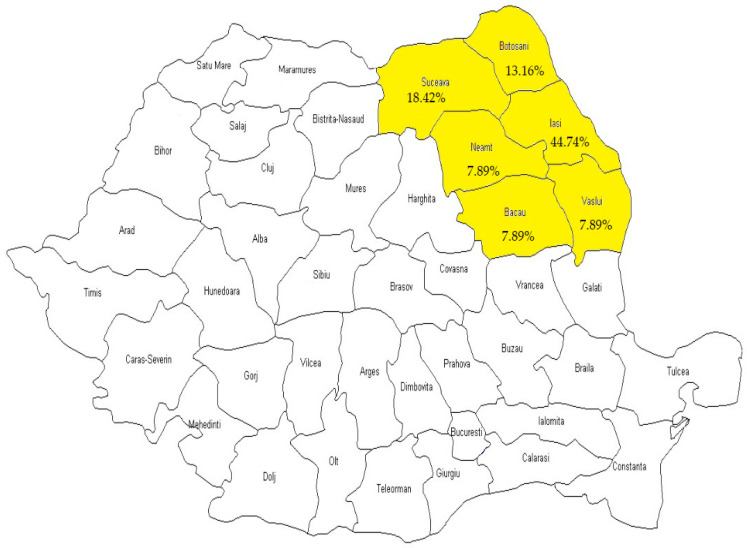
Romania’s northeast counties’ caseload distribution. The yellow color refers exactly to the Romania’s northeast counties.

**Figure 3 medicina-59-01605-f003:**
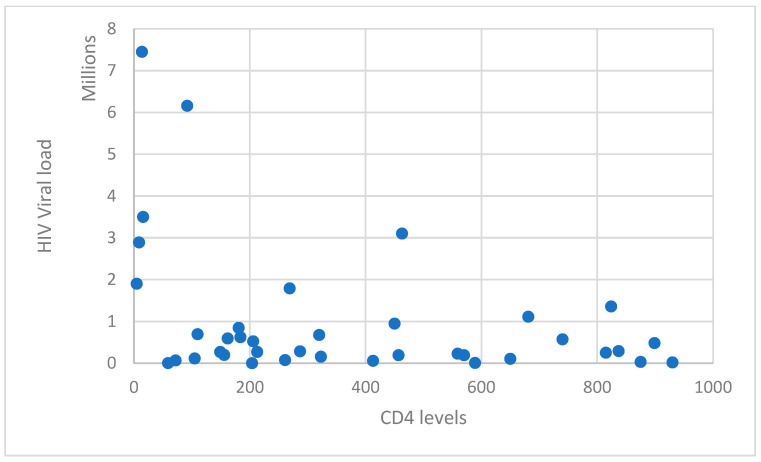
Distribution of cases by HIV viral load and CD4+ T-lymphocyte count.

**Table 1 medicina-59-01605-t001:** Age distribution of HIV/AIDS cases.

Age (Years)	n (38)	%
0–19	1	2.63
20–29	1	2.63
30–39	28	73.68
40–49	4	10.53
50–59	2	5.26
Over 60	2	5.26

**Table 2 medicina-59-01605-t002:** Case distribution according to sex and CD4+ T-lymphocyte count at time of diagnosis.

CD4+ T-lymphocytes Level *p* = 0.137	Male		Female		Total	
	n (21)	%	n (17)	%	n (38)	%
0–199 cells/μL	7	18.42	7	18.42	14	36.84
200–499 cells/μL	8	21.05	4	10.53	12	31.58
>500 cells/μL	6	15.79	6	15.79	12	31.58

**Table 3 medicina-59-01605-t003:** Distribution of cases according to sex, metabolic disorder, and liver enzyme levels.

Laboratory Marker	Value	Male		Female		Total	
		n (21)	%	n (17)	%	n (38)	%
ALT	normal (5–31 UI/L)	14	36.84	12	31.57	26	68.43
	abnormal (>31 UI/L)	7	18.42	5	13.15	12	31.57
AST	normal (5–31 UI/L)	14	36.84	12	31.57	26	68.43
	abnormal (>31 UI/L)	7	18.42	5	13.15	12	31.57
GGT	normal (7–32 UI/L)	12	31.57	11	28.94	23	60.51
	abnormal (>32 UI/L)	9	23.68	6	15.78	15	39.46
Cholesterol	normal (122–200 mg/dL)	11	28.94	12	31.57	23	60.54
	abnormal (>200 mg/dL)	10	26.31	5	13.15	15	39.46
HDL-COL	normal (40–66 mg/dL)	12	31.57	14	36.84	26	68.43
	abnormal (>66 mg/dL)	9	23.68	3	7.89	12	31.57
LDL-COL	normal (30–159 mg/dL)	11	28.94	12	31.57	23	60.54
	abnormal (>159 mg/dL)	10	26.31	5	13.15	15	39.46
Triglycerides	normal (30–150 mg/dL)	13	34.21	10	26.31	23	60.54
	abnormal (>150 mg/dL)	8	21.05	7	18.42	15	39.46

**Table 4 medicina-59-01605-t004:** Opportunistic infections distribution in research sample according to CDC stage.

	Stage 1		Stage 2		Stage 3		Total	
HIV/AIDS Status	n (12)	%	n (12)	%	n (14)	%	n (38)	%
No opportunistic infections	9	23.68	6	15.79	2	5.26	17	44.74
Opportunistic infections	3	7.89	6	15.79	12	31.58	21	55.26

**Table 5 medicina-59-01605-t005:** HIV/AIDS cases by co-infections distribution.

Co-Infections	Men		Women		Total	
*p* = 0.048	n (12)	%	n (7)	%	n (19)	%
HBV	5	13.16	3	7.89	8	21.05
HCV	2	5.26	0	0	2	5.26
Toxoplasmosis	1	2.63	1	2.63	2	5.26
CMV	4	10.53	3	7.89	7	18.42

**Table 6 medicina-59-01605-t006:** Type of ophthalmologic diseases found in our study group.

Type of Disease	n (38)	%
Congenital eye disease	17	44.73
Ophthalmological disease	21	55.26

**Table 7 medicina-59-01605-t007:** Ophthalmologic diseases found in our study patients.

Type of Ophthalmological Disease	n (38)	%
CMV Retinitis	8	21.05
Myopia	7	18.42
Convergent strabismus	3	7.89
Astigmatism	4	10.52
Cataract	5	13.15
Others	11	28.94

**Table 8 medicina-59-01605-t008:** HIV/AIDS cases by ART regimen.

ART Regimen	n (38)	%
Integrase inhibitors+ 2 NNRTI	9	23.68
Protease inhibitors+ 2 NNRTI	17	44.73
NRTI+ 2 NNRTI	10	26.31
Other	2	5.26

**Table 9 medicina-59-01605-t009:** Distribution by sex and the level of CD4+ T-lymphocytes after a month of ART.

CD4 Levels *p* = 0.053	Male		Female		Total	
	n (21)	%	n (17)	%	n (38)	%
0–200 cells/μL	4	10.53	4	10.53	8	21.05
200–499 cells/μL	9	23.68	6	15.79	15	39.47
>500 cells/μL	8	21.05	7	18.42	15	39.47

**Table 10 medicina-59-01605-t010:** HIV viral loads status distribution at baseline and one month following ART.

HIV Viral Load (*p* < 0.05)	Initial Assessment		One Month after ART	
	n (38)	%	n (38)	%
Undetectable (<40 copies/mL)	9	23.68	21	55.26
Detectable (>40 copies/mL)	29	76.32	17	44.74

## Data Availability

All data generated or analyzed during this study are included in this published article.

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
