# Peer review of "Ophthalmological Manifestations in People with HIV from Northeastern Romania"

_medicina, 2023, doi:10.3390/medicina59091605_

Round 1
Reviewer 1 Report
This is a retrospective study of the ophthalmological diseases among people with HIV in Iasi, Romania.
One major limitation of this study is that it only includes patients who were admitted to the hospital, which can skew the results. My specific comments/suggestions are as follows:
1. Suggest to change the phrases "HIV positive" and "HIV-infected" into "people with HIV". This is a more preferred term and less stigmatizing. Please change the title and the text accordingly.
2. Abstract: remove "terrible disease" towards the end of the paragraph as it can be offensive
3. Include standard deviation after average/mean throughout the abstract and Results.
4. Line 67: include citation to the statement: "In contrast to adults with a lower incidence of CMV retinitis, children are more prone..."
5. Include in the Background and Discussion that people with HIV who have CD4 < 50 cells are recommended to have an ophthalmological exam, based on the Infectious Disease Society of America recommendations
6. Line 92: "Before beginning antiretroviral medication, ocular symptoms should be ruled out..." This is NOT accurate. It is now recommended to perform Rapid Start of antiretroviral therapy. The only exclusion criteria are central nervous system infections, such as cryptococcosis. Please provide appropriate citation; alternatively, remove or rephrase this statement.
7. Line 95: "all HIV patients should get a baseline ophtha exam..." -- please provide citation as to what guideline states this
8. Methods: This is of major concern for me:
8a. How many HIV patients in total were admitted to the hospital from Jan 1991 to December 2022? It is hard to believe that only N=38 had eye manifestations throughout this period spanning over 20 years.
8b. How was the review of records performed? Was there an electronic review or were there paper charts?
8c. Among the 38 patients, how was it ascertained that the patients were unique and no patients were admitted/counted twice?
8d. What are the reasons for being admitted to the hospital? Were they all hospitalized? If so, were the eye problems incidental findings (such as myopia), were they from admission, or from self-report?
9. Line 179: clarify "medium level education"
10. Some tables or figures are redundant/not necessary. That is, those that have been mentioned in the text did not have to be presented again as tables. For example, Figure 3, Table 5, Table 11 can be removed
11. Table 6: Indicate the actual cut-off values under 'Value"
12. At the top of the tables, indicate the number of patients after n when appropriate
13. Table 8. The p-value of 0.048 in column 1 header appears misplaced. How was this calculated?
14. Table 14: It is unclear how some patients were undetectable at initial assessment. Were they already on ARVs prior to hospitalization?
15. Limitations of the study have to be mentioned towards the end of the Discussion. This includes small sample size and the retrospective nature of the study, among others.
16. In the Methods section, please provide a description of the hospital (How many beds, is it tertiary referral hospital, etc).
17. 'Drug users' should be changed to 'people who use drugs in Line 343.
English language editing needed for clarity.
Some words need to be changed for inclusivity (see above)
Author Response
Dear Reviewer,
First of all, we would like to express our thanks, the comments regarding the manuscript were very useful, helping to enrich the transmission of information and to add value to our research.
According to the your suggestions we added more data to support our observational retrospective research and express more clearly our goal and results obtained in North-Eastern Romania regarding people with HIV infection.
We adjusted the body text of the manuscript, with modifications in all the sections: introduction, material and methods, results.
We changed the phrases to “people with HIV”, accordingly.
We also enriched the methodology with the standard deviation as suggested.
We added new data in concordance with the Infectious Disease Society of America.
We added data regarding how the review of the records was performed and how each case was unique. We also explained the reason the patients were admitted to the hospital, mainly for routine check-ups.
We reconsidered some of the tables and figures, and modified them accordingly.
We added cut-off values in some of the tables.
Some of the patients, were initially undetectable because they were previously treated in another HIV/AIDS Center.
As well, we mentioned the limitation of the study and we provided a general description of the hospital.
Reviewer 2 Report
Please describe with more detail examples of the ocular findings in congenital HIV versus acquired HIV infections.
Please separate those two groups in the table so that the reader can know exactly if this is a result of congenital HIV versus acquired form
It is unusual that the highest number of patients you have have congenital HIV, since in most centers the truth is that acquired HIV is more prevalent. Is there a reason you suspect that your sample shows that the most common form of HIV is the congenital one?
There are some results presented regarding the metabolic panels and liver panels that do not correlate to the ocular findings, how is that relevant in your study, if only presented and not discussed later.
Please try to alternate between connecting words to make the text more coherent. Also please try to abide to one tens, there are some places where past and present tenses are used in consecutive sentences.
Author Response
Dear Reviewer,
First of all, we would like to express our thanks, the comments regarding the manuscript were very useful, helping to enrich the transmission of information and to add value to our research.
According to the suggestions we added more data to support our observational retrospective research and express more clearly our goal and results obtained in North-Eastern Romania regarding people with HIV infection.
We adjusted the body text of the manuscript, with modifications in all the sections: introduction, material and methods, results.
We described the findings in congenital eye disease and acquired ophthalmological manifestations in individuals with HIV. The congenital and acquired findings refers to the ocular manifestations, not the HIV infection. We don’t have available data to confirm the way of HIV infection transmission.
Regarding the metabolic and liver panels, further studies will focus more on the impact of the enzymes on the eye manifestations in people with HIV or AIDS.